# AdPO: Enhancing the Adversarial Robustness of Large Vision-Language Models with Preference Optimization

## Abstract

Large Vision-Language Models (LVLMs), such as GPT-4 and LLaVA, have recently witnessed remarkable advancements and are increasingly being deployed in real-world applications. However, inheriting the sensitivity of visual neural networks, LVLMs remain vulnerable to adversarial attacks, which can result in erroneous or malicious outputs. While existing efforts utilize adversarial fine-tuning to enhance robustness, they often suffer from performance degradation on clean inputs. In this paper, we proposes AdPO, a novel adversarial defense strategy for LVLMs based on preference optimization. Preference optimization methods, such as DPO and RLHF, have been widely used to align large language models (LLMs) with human values and preferences. For the first time, we reframe adversarial training as a preference optimization problem, aiming to enhance the model's preference for generating normal outputs on clean inputs while rejecting the potential misleading outputs for adversarial examples. Notably, AdPO achieves this by solely modifying the image encoder, e.g., CLIP ViT, resulting in superior robustness across a range of downstream tasks (including LVLMs and zero-shot classification). Our comprehensive experimental validation confirms the efficacy of the proposed AdPO, which outperforms prior state-of-the-art methods.

## 1 Introduction

The emergence of large vision-language models (LVLMs) has substantially propelled the development of general artificial intelligence, attracting considerable attention from the research community (Yin et al., 2023; Cui et al., 2024; Liu et al., 2024b). These models generally consist of two key components: visual modules and Large Language Models (LLMs) (Zhao et al., 2023a). The visual modules, frequently utilizing pre-trained image encoders like CLIP's ViT (Radford et al., 2021), are responsible for extracting salient visual features from images and projecting them onto the input space of the language model. This alignment facilitates the next-token prediction in an autoregressive manner within the framework of the language model. Cutting-edge LVLMs, such as GPT-4 (OpenAI et al., 2024), LLaVA (Liu et al., 2023b), and OpenFlamingo (Awadalla et al., 2023), have demonstrated outstanding capabilities in understanding and reasoning with both visual and textual information. These models have delivered exceptional performance across a broad range of tasks, such as image captioning (Dai et al., 2023; Nguyen et al., 2023), visual question answering (Liu et al., 2023b), and text recognition (Liu et al., 2024a; Li et al., 2023d).

Given their transformative potential for multimodal learning and understanding, LVLMs are positioned for deployment across a growing range of real-world applications. However, this widespread deployment introduces significant security concerns, as malicious attacker could manipulate LVLMs into generating undesirable content and hallucinations (Schlarmann & Hein, 2023; Shayegani et al., 2024). Consequently, it is imperative to rigorously test and improve the robustness of these models prior to deployment. Recent research has identified a critical vulnerability in LVLMs to adversarial attacks targeting both textual and visual inputs (Zhao et al., 2023b). Notably, the continuous nature of the visual modality renders it more susceptible to manipulation via numerical optimization techniques (Wang et al., 2024b; Carlini et al., 2023; Qi et al., 2024b; Luo et al., 2024). Researchers have demonstrated both targeted and untargeted attacks by introducing imperceptible noise into images, which consequently alters the model's interpretation and output.

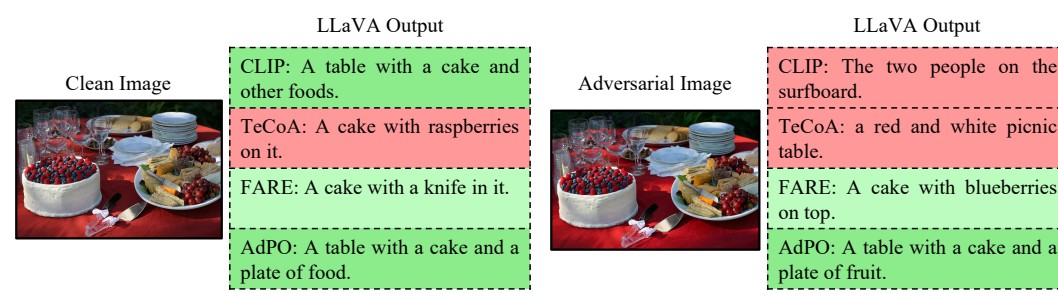

Figure 1: Illustration of f adversarial attacks with $\epsilon = 4/255$ on LLaVA using different CLIP models. The original model can produce accurate outputs on clean images, but it makes significant errors when faced with adversarial attacks. Although the adversarially trained versions, TeCoA and FARE, have better adversarial robustness, they still tend to hallucinate or fail to fully comprehend the image. Comparatively, our AdPO exhibits strong performance on both clean and adversarially altered images.

To improve the adversarial robustness of LVLMs, existing efforts focus on fine-tuning primarily the image encoder. For example, TeCoA utilizes a text-guided contrastive adversarial training loss, supervising the alignment of text embeddings with adversarial visual features on a limited training dataset (Mao et al., 2023). FARE proposes an unsupervised adversarial fine-tuning scheme to eliminate the dependence on labeled training datasets (Schlarmann et al., 2024). Although these methods have achieved advancements in improving the robustness of CLIP models, they continue to suffer from performance degradation on downstream tasks, including LVLMs and zero-shot classification. As shown in Figure 1, TeCoA generates severe hallucinations with clean samples, whereas FARE tends to lose its fine-grained comprehension of the image.

Inspired by the significant success of preference optimization in the LLM community (Wang et al., 2024e; Ouyang et al., 2022), we find that applying preference optimization to adversarial training is highly promising, given the alignment between their objectives. More specifically, adversarial training aims to enhance model robustness against adversarial attacks while preserving performance on clean data. Preference optimization, such as DPO (Rafailov et al., 2023), aligns LLMs with human values by increasing the probability of preferred outputs while decreasing the likelihood of non-preferred ones. Leveraging this insight, we propose AdPO, a novel **A**dversarial **d**efense strategy based on **P**reference **O**ptimization, which enables LVLMs to generate correct outputs from clean image inputs while rejecting misleading outputs from adversarial images.

However, applying DPO to adversarial training presents unique challenges. In comparison to standard offline DPO, we introduce two key improvements: (1) To remove the reliance on image annotations, we adapt DPO to an online setting. During training, the policy model generates interpretations for both clean and adversarial images, which serve as sources for positive and negative samples. This process is referred to as **preferred image optimization**. (2) Multimodal preference optimization may face an *unconditional preference* issue, where the learning process may neglect image conditions (Wang et al., 2024a). To address this issue, we introduce supplementary **adversarial image optimization** to further improve the adversarial robustness of LVLMs. To ensure consistency with previous research, we confine our adversarial training to adjusting only the parameters of CLIP's ViT on the ImageNet dataset (Deng et al., 2009). Extensive experimental results, including those on LVLMs and zero-shot classification, demonstrate that our proposed AdPO achieves a more robust image encoder, with minimal impact on clean inputs and even shows improvements in certain tasks. These outcomes not only validate the effectiveness of our approach but also expand the potential applications of preference optimization techniques beyond their original scope in language models.

In summary, our contributions can be summarized as follows:

- We introduce AdPO (Adversarial defense based on Preference Optimization), which, to the best of our knowledge, is the first attempt to explore the application of preference optimization for adversarial training.

- We propose the dual strategy of preferred image optimization and adversarial image optimization to maintain the model's clean performance while enhancing its adversarial robustness.

- Extensive experiments show that our method achieves state-of-the-art results in improving the adversarial robustness of LVLMs while maintaining the original performance as much as possible.

## 2 RELATED WORK

In this section, we primarily review the related studies on large vision-language models, adversarial attacks, adversarial defenses, and preference optimization methods.

**Large Vision-Language Models.** Recently, large multimodal models have emerged, including LLaVA 1.5 (Liu et al., 2023a), OpenFlamingo (OF) (Awadalla et al., 2023), BLIP-2 (Li et al., 2023b), MiniGPT-4 (Zhu et al., 2024), Otter (Li et al., 2023a), mPLUG-Owl (Ye et al., 2023), Qwen-VL (Bai et al., 2023), MiniCPM-V (Yao et al., 2024), DeepSeek-VL (Lu et al., 2024), InternVL (Chen et al., 2024), and Idefics2 (Laurençon et al., 2024). These models typically use pre-trained image encoders (e.g., CLIP or SigCLIP) to extract image features, which are then aligned with text embedding spaces (Radford et al., 2021; Zhai et al., 2023). The visual and textual embeddings are then fed into LLMs for autoregressive generation. This approach allows the model to simultaneously understand and generate content related to both images and text. To mitigate computational load, a practical strategy is to freeze the image encoder and train only the projection layer, which not only simplifies the training process but also enhances efficiency (Liu et al., 2023b; Awadalla et al., 2023). Therefore, image encoders can significantly impact the performance of LVLMs, receiving significant attention from the multimodal community (Cao et al., 2023). We focus on the performance evaluation of LLaVA-1.5 and OF, as both use CLIP ViT-L/14 (Radford et al., 2021) as their image encoder.

**Adversarial attacks.** The vulnerability of visual neural network models to adversarial attacks is well-established and has been extensively investigated (Szegedy et al., 2014; Goodfellow et al., 2015; Madry et al., 2018; Brown et al., 2017; Zhang et al., 2023). By introducing carefully crafted noise into images, adversaries can cause the victim model to generate incorrect outputs with high confidence. Capitalizing on this vulnerability, recent studies have shown that LVLMs are also vulnerable to attacks targeting visual inputs Schlarmann & Hein (2023); Shayegani et al. (2024); Luo et al. (2024); Gao et al. (2024); Dong et al. (2023b). Zhao et al. (2023b) showed that transferable black-box attacks could be generated using text-to-image models. Carlini et al. (2023) demonstrated how adding adversarial noise to images can circumvent safety constraints of LLMs. Qi et al. (2024a) explored how adversarial attacks embedding deceptive information into images can mislead LVLMs and deceive users. The widespread deployment of LVLMs has raised urgent security concerns due to the threat of adversarial attacks.

**Adversarial defenses.** Adversarial defenses in machine learning safeguard models from malicious inputs to ensure their integrity and reliability, especially in security-sensitive contexts (Madry et al., 2018; Fares et al., 2024; Papernot et al., 2016; Meng & Chen, 2017; Zhou & Patel, 2022). Adversarial training is a foundational method for enhancing a model's inherent robustness by integrating adversarial examples into the training dataset Kurakin et al. (2017b); Tramèr et al. (2018); Dong et al. (2023a). In the multimodal domain, TeCoA improves the adversarial robustness of CLIP's image encoder through text-guided contrastive adversarial training while preserving some of CLIP's zero-shot classification capabilities (Mao et al., 2023). FARE employs unsupervised training by minimizing the distance between adversarial image features and clean image features, maintaining impressive performance on LVLMs (Schlarmann et al., 2024). However, this straightforward adversarial training approach often fails to prevent performance degradation on clean samples. Unlike these fine-tuning strategies, we are the first to frame adversarial training as a preference optimization problem, integrating both clean and adversarial images into the training process to improve robustness while maintaining clean performance.

**Preference optimization.** Preference optimization has emerged as a novel training paradigm for aligning LLMs with human values and has garnered significant attention in recent research (Ouali et al., 2024; Yu et al., 2023; 2024; Wang et al., 2024a;c). Reinforcement Learning from Human Feedback (RLHF) utilizes human preferences as a reward model and applies reinforcement learn-

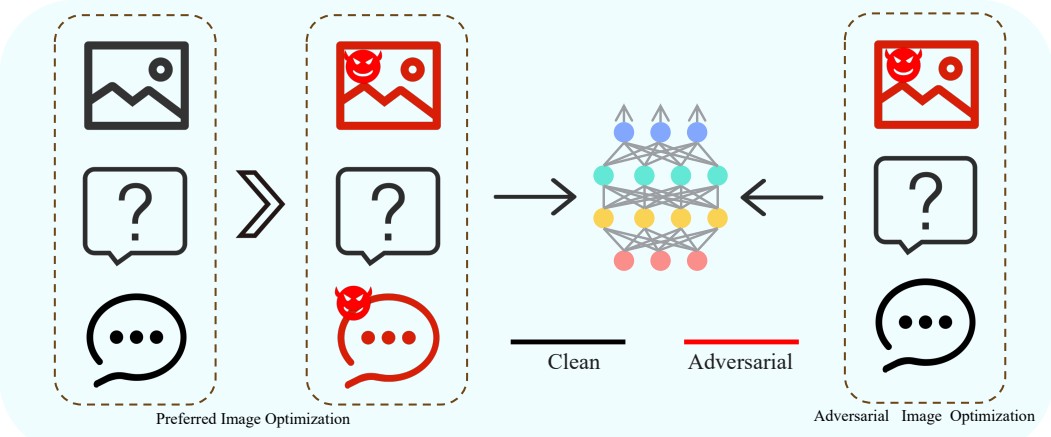

Figure 2: The architecture of our proposed AdPO. AdPO mainly consists of two parts: (**left**) preferred image optimization and (**right**) adversarial image optimization. Preferred image optimization incorporates both clean and adversarial images into adversarial training while maintaining the model's performance on clean inputs, and adversarial image optimization can significantly enhance the model's adversarial robustness.

ing to guide model training (Bai et al., 2022; Ouyang et al., 2022) Direct Preference Optimization (DPO) streamlines the training process by increasing the log probability of preferred samples while reducing that of non-preferred samples, enabling broader applications (Rafailov et al., 2023). Subsequent advancements, such as StepDPO (Lai et al., 2024), SmiPO (Meng et al., 2024), and IPO (Azar et al., 2024), have further improved DPO's performance. Considering its stability and efficiency in training, we also adopt DPO for adversarial training of LVLMs in this work.

## 3 METHOD

This section provides a detailed introduction to our AdPO, with its overall framework illustrated in Figure 2. First, Section 3.1 outlines the basics of the DPO algorithm, and Section 3.2 discusses adversarial example generation, which forms the preference sample pairs required for DPO. Sections 3.3 and 3.4 introduce preferred image optimization and adversarial image optimization, respectively.

### 3.1 PRELIMINARIES

DPO has emerged as a prominent method in the domain of offline preference optimization. This method provides a novel framework for optimizing language models in accordance with human preferences. In a typical setup, given an input $x$ and an output text $y$, a language model (i.e., policy model) $\pi_\theta$ generates a conditional distribution $\pi_\theta(y|x)$. Unlike RLHF, which employs an explicit reward model, DPO reformulates the reward function using a closed-form expression with respect to the optimal policy. The main objective of DPO is to maximize the expected reward of the outputs generated by this policy, with the reward function defined as $r(x, y)$:

$$r(x,y) = \beta \log \frac{\pi_\theta(y|x)}{\pi_{\text{ref}}(y|x)} + \beta \log Z(x), \tag{1}$$

where $\beta$ is a constant, $\pi_{\text{ref}}$ is the reference policy model (identical to the original $\pi_\theta$), and $Z(x)$ is the partition function.

Given a preference dataset $\mathcal{D} = \{x, y_w, y_l\}$, where $y_w$ and $y_l$ represent the winning and losing responses respectively, DPO employs a Bradley-Terry model (Bradley & Terry, 1952) to express the probability for each preference pair:

$$p(y_w \succ y_l) = \sigma(r(x, y_w) - r(x, y_l)), \tag{2}$$

where $\sigma(\cdot)$ is typically defined as a sigmoid function. The key innovation of DPO is its formulation of the likelihood of preference data using the policy model, as opposed to relying on an explicit reward model. This leads to the formulation of the DPO objective:

$$\mathcal{L}_{\text{DPO}}(\pi_\theta; \pi_{\text{ref}}) = -\mathbb{E}_{(x,y_w,y_l)\sim\mathcal{D}} \left[ \log \sigma \left( \beta \log \frac{\pi_\theta(y_w|x)}{\pi_{\text{ref}}(y_w|x)} - \beta \log \frac{\pi_\theta(y_l|x)}{\pi_{\text{ref}}(y_l|x)} \right) \right], \quad (3)$$

This formulation captures the core principles of DPO, providing a robust framework for optimizing language models in alignment with human preferences.

## 3.2 Adversarial Example Generation

In the context of large vision-language models, the input to the model comprises $x = \{x_m, x_t\}$, where $x_m$ denotes the image input and $x_t$ represents the text input. This section outlines the principles behind generating adversarial images.

Adversarial images are generated by introducing small, nearly imperceptible perturbations to original images, with the goal of deceiving machine learning models and inducing incorrect predictions (Szegedy et al., 2014; Goodfellow et al., 2015). Although adversarial images appear nearly identical to the original images to humans, they can drastically alter the model's output, exposing its vulnerability to malicious inputs (Kurakin et al., 2017a). Adversarial attacks can be broadly categorized into targeted and untargeted attacks: targeted attacks compel the model to produce specific outputs (Luo et al., 2024), whereas untargeted attacks merely lead the model to generate incorrect outputs (Wang et al., 2024d; Gao et al., 2024). In this study, we employ untargeted attack methods to generate adversarial images. This approach eliminates reliance on specific labeled datasets, enabling our method to be extended to unseen datasets.

Given an image encoder $\phi$, (e.g., CLIP ViT) and a clean image $x_m$, adversarial examples are generated by optimizing to maximize the discrepancy between the encoded features of the adversarial image and the clean image:

$$x_{adv} = \underset{\|x_{adv}-x_m\|_\infty \leq \varepsilon}{\arg\max} \|\phi(x_{adv}) - \phi_{\text{org}}(x_m)\|_2^2. \quad (4)$$

where $x_{adv}$ is the adversarial image obtained through iterative optimization like PGD (Madry et al., 2018), $\phi_{org}$ is the original image encoder and $\epsilon$ is the image perturbation magnitude. Note that in subsequent adversarial training, the parameters of $\phi$ will be updated.

## 3.3 Preferred Image Optimization

This section primarily outlines the process of constructing pairs of preferred and non-preferred samples from unlabeled image data, a crucial component of the DPO training pipeline.

Given a clean image $x_m$ and its adversarial image $x_{adv}$, we employ an online approach to directly prompt the model (e.g., *"What is the content of the image?"*) to generate interpretations, thereby obtaining the preferred response $y_w$ and the non-preferred response $y_l$. Accordingly, in the setting of multimodal adversarial training, our preferred image optimization can be formulated as:

$$\mathcal{L}_{\text{P}}(\pi_\theta; \pi_{\text{ref}}) = -\mathbb{E}_{(x_m,x_t,y_w,y_l)\sim\mathcal{D}} \left[ \log \sigma \left( \beta \log \frac{\pi_\theta(y_w|x_m,x_t)}{\pi_{\text{ref}}(y_w|x_m,x_t)} - \beta \log \frac{\pi_\theta(y_l|x_{adv},x_t)}{\pi_{\text{ref}}(y_l|x_{adv},x_t)} \right) \right], \quad (5)$$

This straightforward approach presents several advantages. First, it removes the need for data annotation, thus facilitating its application to previously unseen image data. Second, this method resembles semi-supervised learning, especially as LVLMs now possess advanced capabilities, enabling them to incorporate labeled images into their training data. Moreover, allowing the model to generate its own text as labels effectively mitigates distribution shift issues, thus concentrating attention on the adversarial images themselves (Li et al., 2023c).

## 3.4 Adversarial Image Optimization

Although preferred image optimization can maintain the performance of VLMs on clean inputs, it does not significantly enhance adversarial robustness. Recent research indicates that, although multimodal DPO is designed to compute implicit rewards based on all input modalities, it may prioritize

language-only preferences while neglecting image conditions (i.e., unconditional preferences), resulting in suboptimal model performance and increased hallucinations (Wang et al., 2024a).

The issue of unconditional preferences may lead to suboptimal adversarial robustness. To address this, we introduce adversarial image optimization:

$$\mathcal{L}_{\mathrm{A}} = \sum_{t=1}^{T} \log \pi_\theta(y_w^t \mid x_{adv}, x_t^{1:t-1}), \tag{6}$$

where $T$ represents the sequence length of each sample. The objective of AdPO is a combination of preferred image optimization and adversarial image optimization:

$$\mathcal{L}_{\mathrm{AdPO}} = \mathcal{L}_{\mathrm{P}} + \mathcal{L}_{\mathrm{A}}. \tag{7}$$

By leveraging joint optimization, AdPO attains enhanced adversarial robustness while maintaining its performance on clean samples.

## 4 EXPERIMENTS

In this section, we evaluate the performance of AdPO on LVLMs and zero-shot classification through extensive experiments. Although we use the complete LVLMs during adversarial training, we modify only the parameters of their image encoders, enabling the robust image encoder to be directly transferred to other LVLMs. All experiments are conducted on 32 Tesla A100 GPUs.

**Models.** For the LVLM models, we primarily select OpenFlamingo-9B (OF)(Awadalla et al., 2023) and LLaVA 1.5-7B(Liu et al., 2023a), both of which use CLIP's ViT-L/14 as their image encoder (Radford et al., 2021). The two models differ in their language decoders: OF employs MPT-7B (Team et al., 2023), while LLaVA 1.5 uses Vicuna (Chiang et al., 2023). In the subsequent evaluation of OF, we adopt a zero-shot setting, where the model is given textual prompts without any accompanying images (Alayrac et al., 2022). For LLaVA, we employ the default system prompt along with task-specific prompts (Liu et al., 2023b).

**Adversarial training settings.** In AdPO, we leverage LLaVA 1.5 to fine-tune CLIP's ViT model on the ImageNet dataset (Deng et al., 2009). As we adopt an online learning approach, we do not rely on category labels provided by the dataset, only on the images themselves. By optimizing Equation 4, we generate adversarial perturbations for clean images using a 10-step PGD under the $\ell_\infty$ norm. It is widely recognized that employing larger image perturbations during adversarial training can significantly improve adversarial robustness, but it often leads to performance degradation on clean data (Madry et al., 2018). To balance robustness and clean accuracy, we apply two perturbation radii: $\epsilon = 2/255$ and $\epsilon = 4/255$. The resulting robust CLIP image encoders are referred as AdPO[2] and AdPO[4], respectively. We use the AdamW optimizer with a weight decay of 1e-4 and a learning rate of 1e-5. We conduct training for two epochs with a batch size of 128. The preference optimization parameter $\beta$ is set to 0.1.

**Baseline methods.** We compare the performance of AdPO with the original CLIP and two state-of-the-art methods, TeCoA (Mao et al., 2023) and FARE (Schlarmann et al., 2024). TeCoA utilizes supervised contrastive learning with image category labels, while FARE performs unsupervised training at the representation level. To ensure fair comparison, we use adversarial images with the same noise radius for training, denoted as TeCoA[2] and FARE[2] for $\epsilon = 2/255$, and TeCoA[4] and FARE[4] for $\epsilon = 4/255$.

### 4.1 EVALUATION OF UNTARGETED ATTACKS ON LVLMS

In this section, we evaluate the clean and robust performance of AdPO in vision-language tasks by replacing the image encoder of LVLMs with robust versions.

**Attack setup.** We utilize the approach outlined in Schlarmann & Hein (2023) to perform untargeted attacks aimed at degrading the model's performance. Given that attacks on LVLMs often demand more iterations, we employ a 100-step APGD attack (Croce & Hein, 2020), which utilizes ground-truth captions as labels. After each attack, we discard samples with scores below a specified threshold to ensure that computationally expensive attacks are only performed when necessary, following Schlarmann et al. (2024). Further details are provided in the Appendix A.1.

Table 1: Evaluation of the adversarial robustness of large vision-language models with different CLIP models. We evaluate the clean performance and adversarial robustness of various methods across multiple tasks and perturbation sizes. The results indicate that AdPO significantly exceeds our baseline methods, attaining outstanding robustness along with exceptional clean performance. The best results are shown in **bold**.

| VLM | Image Encoder | COCO clean | $\ell_\infty$ 2/255 | 4/255 | Flickr30k clean | $\ell_\infty$ 2/255 | 4/255 | TextVQA clean | $\ell_\infty$ 2/255 | 4/255 | VQAv2 clean | $\ell_\infty$ 2/255 | 4/255 |
|---|---|---|---|---|---|---|---|---|---|---|---|---|---|
| OF-9B | CLIP | 79.7 | 1.5 | 1.1 | 60.1 | 0.7 | 0.4 | 23.8 | 0.0 | 0.0 | 48.5 | 1.8 | 0.0 |
| | TeCoA[2] | 73.5 | 31.5 | 21.2 | 49.5 | 14.1 | 9.5 | 16.6 | 3.5 | 2.1 | 46.2 | 23.5 | 20.5 |
| | FARE[2] | 79.1 | 34.2 | 19.5 | 57.7 | 16.4 | 8.9 | 21.6 | 4.1 | 1.9 | 47.0 | 24.0 | 17.2 |
| | AdPO[2] | **84.7** | **34.6** | **25.5** | **57.9** | **18.8** | **12.3** | **22.3** | **6.5** | **3.3** | **48.1** | **26.3** | **22.8** |
| | TeCoA[4] | 66.9 | 28.5 | 21.6 | 40.9 | 12.0 | 10.3 | 15.4 | 2.1 | 1.8 | 44.8 | 23.6 | 21.3 |
| | FARE[4] | 74.1 | 30.9 | 22.8 | 51.4 | 15.7 | 10.5 | 18.6 | 3.4 | 2.9 | 46.1 | 23.6 | 21.0 |
| | AdPO[4] | **75.2** | **33.3** | **25.9** | **54.6** | **17.2** | **12.7** | **20.5** | **5.2** | **3.3** | **46.7** | **24.4** | **21.3** |
| LLaVA 1.5-7B | CLIP | 115.5 | 4.0 | 3.1 | 77.5 | 1.6 | 1.0 | 37.1 | 0.5 | 0.0 | 74.5 | 2.9 | 0.0 |
| | TeCoA[2] | 98.4 | 44.2 | 30.3 | 57.1 | 23.2 | 15.3 | 24.1 | 12.1 | 8.8 | 66.9 | 33.8 | 21.8 |
| | FARE[2] | 109.9 | 53.6 | 31.0 | 71.1 | 29.5 | 17.5 | 31.9 | 14.7 | 9.1 | 71.7 | **34.9** | 23.0 |
| | AdPO[2] | **118.3** | **65.3** | **43.9** | **75.4** | **32.5** | **20.1** | **32.4** | **17.8** | **10.5** | **72.9** | 34.3 | **23.2** |
| | TeCoA[4] | 88.3 | 50.9 | 35.3 | 48.6 | 27.9 | 19.5 | 20.7 | 12.6 | 9.3 | 63.2 | 41.0 | 31.7 |
| | FARE[4] | 102.4 | 57.1 | 40.9 | 61.6 | 31.4 | 22.8 | 27.6 | 15.8 | **10.9** | 68.3 | 40.7 | 30.5 |
| | AdPO[4] | **111.5** | **67.2** | **49.3** | **67.0** | **35.3** | **25.4** | **32.3** | **16.1** | 10.2 | **70.1** | **42.3** | **32.5** |

**Datasets and metrics.** We utilize a variety of datasets for image captioning tasks, including COCO (Lin et al., 2014) and Flickr30k (Plummer et al., 2015), as well as for visual question answering tasks, such as VQAv2 (Goyal et al., 2017) and TextVQA (Singh et al., 2019). Considering that adversarial attacks are time-consuming and costly, we randomly selected 500 images for evaluation. We employ the CIDEr score (Vedantam et al., 2015) for image captioning and VQA accuracy (Antol et al., 2015) for visual question answering tasks to present our results.

Table 1 summarizes the experimental results. Typically, the original CLIP model achieves optimal clean performance but lacks adversarial robustness, rendering it vulnerable to attacks. When comparing different methods, our AdPO consistently achieves superior clean performance and adversarial robustness compared to baseline methods, emphasizing the significance of including both clean and adversarial images in the training dataset. Across various datasets, our method demonstrates significant improvements in tasks such as COCO image captioning, likely due to the alignment between this task and our adversarial training paradigm, enabling the robust model to potentially outperform the clean model. For different perturbation sizes, $\epsilon = 2/255$ already ensures solid adversarial robustness, while larger perturbations still preserve more clean performance. AdPO[4] exhibits stronger robustness compared to AdPO[2], but at the cost of some clean performance.

## 4.2 Evaluation of Targeted Attacks on LVLMs

In contrast to the untargeted attacks discussed in Section 4.1, targeted attacks on LVLMs pose a significantly greater threat. Targeted attacks aim to compel the model to produce specific outputs, with the added noise in the image remaining imperceptible to the user. Through image manipulation, attackers can circumvent the model's security mechanisms, leading it to generate malicious content (Carlini et al., 2023; Niu et al., 2024; Qi et al., 2024b). Additionally, attackers can embed phishing links into images through adversarial attacks to deceive users (Bagdasaryan et al., 2023). In this section, we examine the robustness of substituting the CLIP encoder in LLaVA with our adversarially robust variant.

**Attack setup.** We perform targeted attack experiments on LLaVA 1.5-7B, using the attack success rate (ASR) as the primary evaluation metric. A sample is deemed successfully attacked if the model's

Table 2: Quantitative evaluation of targeted attacks at $\epsilon = 4/255$ radii. We assess the Attack Success Rate (ASR) for each setup.

| Target | CLIP | TeCoA$^2$ | FARE$^2$ | AdPO$^2$ | TeCoA$^4$ | FARE$^4$ | AdPO$^4$ |
|---|---|---|---|---|---|---|---|
| `A group of people are playing...` | 20/20 | 1/20 | 1/20 | 0/20 | 0/20 | 0/20 | 0/20 |
| `A group of people are flying...` | 20/20 | 1/20 | 1/20 | 0/20 | 0/20 | 0/20 | 0/20 |
| `The pizza on the table...` | 20/20 | 2/20 | 0/20 | 0/20 | 0/20 | 0/20 | 0/20 |
| `An earthquake is about...` | 20/20 | 2/20 | 1/20 | 1/20 | 0/20 | 0/20 | 0/20 |
| `This patient needs the best...` | 20/20 | 0/20 | 0/20 | 0/20 | 0/20 | 0/20 | 0/20 |
| **Mean ASR:** | 100% | 4% | 3% | 1% | **0%** | **0%** | **0%** |

Table 3: Evaluation of clean and adversarial performance on image classification datasets using the CLIP model. We primarily evaluate the performance of the original CLIP model and its adversarially trained versions when faced with clean samples and adversarial samples with a noise $4/255$. Detailed descriptions of the dataset are provided in the appendix.

| Eval. | Image Encoder | CalTech | Cars | CIFAR10 | CIFAR100 | DTD | EuroSAT | FGVC | Flowers | ImageNet-R | ImageNet-S | PCAM |
|---|---|---|---|---|---|---|---|---|---|---|---|---|
| Clean | CLIP | 83.3 | 77.9 | 95.2 | 71.1 | 55.2 | 62.6 | 31.8 | 79.2 | 87.9 | 59.6 | 52.0 |
| | TeCoA$^2$ | 80.7 | 50.1 | 87.5 | 60.7 | 44.4 | 26.1 | 14.0 | 51.8 | 80.1 | 58.4 | 49.9 |
| | FARE$^2$ | 84.8 | 70.5 | 89.5 | 69.1 | 50.0 | 25.4 | 26.7 | 70.6 | 85.5 | 59.7 | 50.0 |
| | AdPO$^2$ | 85.1 | 72.8 | 91.2 | 69.5 | 53.1 | 35.3 | 25.9 | 74.4 | 87.5 | 59.6 | 50.7 |
| | TeCoA$^4$ | 78.4 | 37.9 | 79.6 | 50.3 | 38.0 | 22.5 | 11.8 | 38.4 | 74.3 | 54.2 | 50.0 |
| | FARE$^4$ | 84.7 | 63.8 | 77.7 | 56.5 | 43.8 | 18.3 | 22.0 | 58.1 | 80.2 | 56.7 | 50.0 |
| | AdPO$^4$ | 84.9 | 65.8 | 80.2 | 56.6 | 44.5 | 21.7 | 21.4 | 58.5 | 82.9 | 57.8 | 49.9 |
| $\epsilon = 4/255$ | CLIP | 0.0 | 0.0 | 0.0 | 0.0 | 0.0 | 0.0 | 0.0 | 0.0 | 0.0 | 0.0 | 0.0 |
| | TeCoA$^2$ | 57.4 | 6.5 | 31.0 | 17.8 | 14.7 | 7.7 | 1.1 | 9.8 | 36.7 | 32.8 | 16.0 |
| | FARE$^2$ | 46.6 | 4.8 | 25.9 | 13.9 | 11.7 | 0.5 | 0.6 | 7.1 | 25.6 | 22.5 | 17.2 |
| | AdPO$^2$ | 55.3 | 5.8 | 28.7 | 17.5 | 13.6 | 5.7 | 1.0 | 8.7 | 33.4 | 33.1 | 15.8 |
| | TeCoA$^4$ | 60.9 | 8.4 | 37.1 | 21.5 | 16.4 | 6.6 | 2.1 | 12.4 | 41.9 | 34.2 | 44.0 |
| | FARE$^4$ | 64.1 | 12.7 | 34.6 | 20.2 | 17.3 | 11.1 | 2.6 | 12.5 | 40.6 | 30.9 | 50.2 |
| | AdPO$^4$ | 66.8 | 13.6 | 36.9 | 21.7 | 17.9 | 9.2 | 2.6 | 12.7 | 42.3 | 33.3 | 49.7 |

output contains the target string. Targeted attacks on LVLMs generally require more iterations, prompting us to execute APGD attacks for 10,000 iterations. Given that larger image perturbations pose more significant threats, we employ $\ell_\infty$ threat models with a radius of $\epsilon = 4/255$. We test five target strings, sampling 20 images for each string.

The quantitative evaluation results are presented in Table 2. The attack success rate for the clean version of the CLIP model reaches 100%, underscoring the vulnerability of current vision-language models to visual input and the substantial security risks posed. TeCoA$^2$, FARE$^2$, and AdPO$^2$ demonstrate varying degrees of adversarial robustness, even when subjected to higher levels of adversarial noise. By comparison, the $\epsilon = 4/255$ versions exhibit significantly higher levels of adversarial robustness. Additional details are provided in Appendix A.2.

### 4.3 EVALUATION OF ZERO-SHOT CLASSIFICATION

In this section, we assess the zero-shot classification performance of the robust CLIP image encoder, following the methods of Mao et al. (2023) and Schlarmann et al. (2024). CLIP's zero-shot classi-

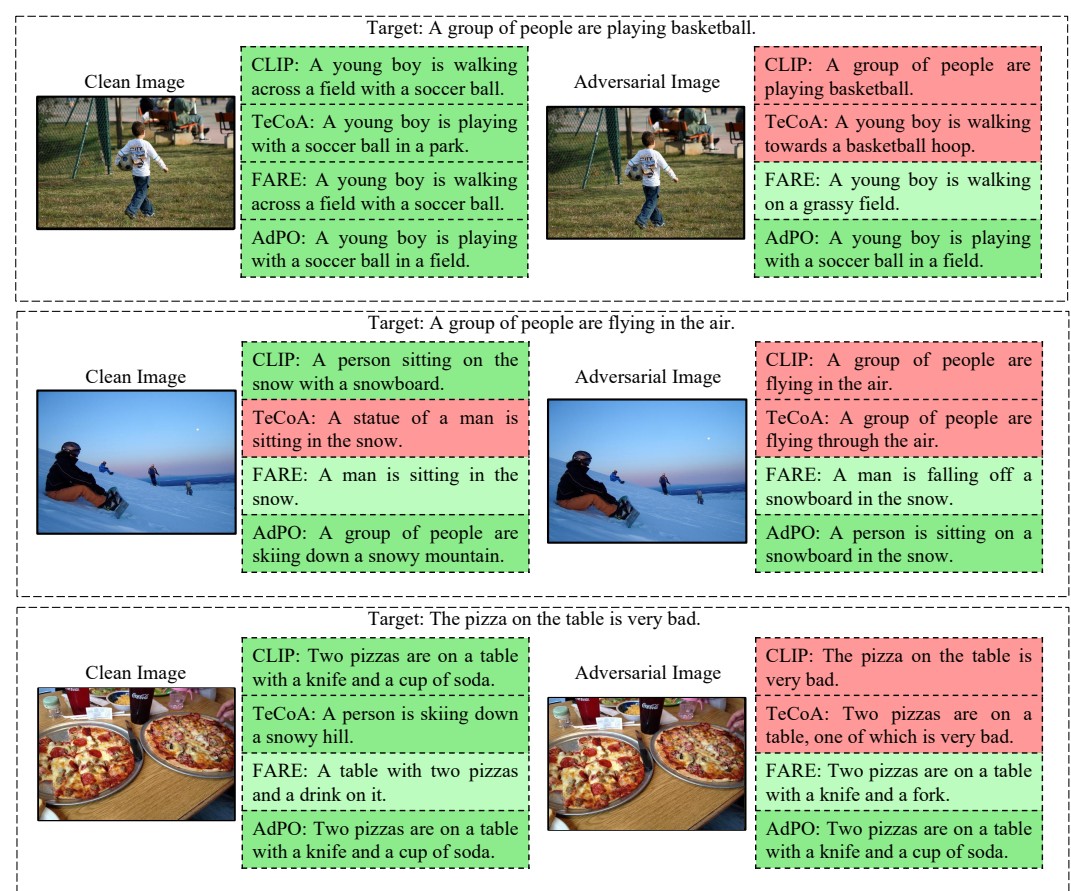

Figure 3: Qualitative assessment of targeted attacks on LLaVA. (Left) When encountering clean images, CoTeA may exhibit noticeable errors, which is undesirable in adversarial defense, while FARE and AdPO demonstrate better clean performance. (Right) When faced with adversarial images, the original CLIP version of LLaVA is easily compromised, FARE shows some adversarial robustness but loses more details or makes subtle errors, whereas AdPO performs better.

fication simultaneously trains both visual and text encoders, enabling the model to project images and textual descriptions into a shared semantic space. For classification, there is no requirement for a specially labeled dataset for each category; instead, CLIP computes the similarity between images and the textual descriptions of categories to classify images into the most relevant category.

**Attack setup.** To assess the adversarial robustness of the models, we utilize the initial two components of AutoAttack (Croce & Hein, 2020), specifically APGD with cross-entropy loss and APGD with DLR loss, both executed over 100 iterations. In alignment with AutoAttack, we adopt the targeted version of the DLR loss, differing from Mao et al. (2023), where the less effective untargeted variant was applied. We perform the evaluation with a more powerful attack ($\epsilon = 4/255$) in this section and present the $\epsilon = 2/255$ results in Appendix A.3.

As demonstrated in Table 3, similar to evaluations on vision-language tasks, the original CLIP typically achieved the best clean performance but displayed minimal adversarial robustness. Adversarial attacks on the clean CLIP achieved a 100% attack success rate, further confirming CLIP's inherent vulnerability, which introduces several weaknesses in LVLMs. After adversarial training, CLIP exhibits some performance decline on clean samples, but its adversarial robustness significantly improves. In contrast, the AdPO models, particularly AdPO[2], demonstrate substantially higher accuracy on clean data while still preserving robustness.

## 4.4 QUANTITATIVE EVALUATION

In addition to quantitative experimental evaluations, we also present a qualitative comparison of different defense methods in this section.

As depicted in Figure 3, the LLaVA model, using the original CLIP as the encoder, provides the most accurate and detailed understanding of clean images. However, when faced with adversarial images generated by targeted attacks, they are completely vulnerable to successful attacks. TeCoA fails to exhibit robust performance against both clean and adversarial images, whereas FARE experiences a loss of detail or minor errors in image understanding, ultimately falling short of optimal performance. In the absence of adversarial defenses, LLaVA is susceptible to manipulation, resulting in biased outputs that can mislead users and have detrimental effects. Therefore, it is imperative to enhance the model's adversarial robustness.

## 4.5 ABLATION STUDY

In this section, we mainly discuss the impact of preferred image optimization (PIO) and adversarial image optimization (AIO) on the final performance.

We use the setup in Section 4.1 to perform untargeted attacks to evaluate the effectiveness of methods trained with a single optimization approach on the COCO dataset, with experimental results shown in Table 4. PIO retains more of the model's clean performance, but only shows a small amount of adversarial robustness. AIO somewhat weakens the model's clean performance, but significantly improves its adversarial robustness. It can also be observed that PIO contributes to enhancing adversarial robustness, indicating the potential of preference optimization in improving adversarial robustness.

Table 4: Ablation study of preferred image optimization and adversarial image optimization.

| Metric | Clean | $2/255$ | $4/255$ |
|--------|-------|---------|---------|
| PIO    | 119.5 | 35.5    | 29.7    |
| AIO    | 102.4 | 65.8    | 42.1    |
| AdPO   | 118.3 | 65.3    | 49.9    |

## 5 CONCLUSION

We propose AdPO, the first adversarial defense strategy based on preference optimization. The core idea of preference optimization methods, represented by DPO, is to learn both positive and negative samples simultaneously and optimize the model to better align with user preferences or goals. This is achieved by comparing the differences between positive and negative samples, clarifying the direction in which the model should be optimized. Unlike previous adversarial fine-tuning methods, which typically only impose single-target constraints to improve adversarial robustness, leading to a loss of clean performance. In contrast, AdPO explicitly optimizes two objectives: improving adversarial robustness while maintaining proper understanding of clean images. Both quantitative and qualitative experimental analyses demonstrate the superiority of our proposed method, offering a new perspective for future adversarial defense research. Considering that preference optimization is gaining increasing attention in academia, introducing more refined methods into the adversarial defense field could lead to better outcomes.

**Limitations.** Although this paper primarily focuses on LVLMs using CLIP ViT as the encoder, other types of models are equally applicable. Considering the computational resources and alignment with previous work, we only adjusted the parameters of the image encoder, but full tuning may yield better results. Carefully crafted malicious prompts also pose significant security risks to the model, and future work needs to address threats from both image and text inputs. While we have performed a significant amount of evaluation, it is clear that evaluating the adversarial robustness of LVLMs in real-world settings is also essential.

## REPRODUCIBILITY

To ensure the reproducibility of our method, we provide a detailed description of our experimental setup in the experiment and appendix sections. The training datasets, evaluation datasets, and involved models are all openly available and accessible.

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

# A APPENDIX

## A.1 UNTARGETED ATTACK DETAILS

We provide a comprehensive description of the attack pipeline utilized for the untargeted adversarial evaluation of LVLMs in Section 4.1.

For the captioning tasks COCO and Flickr30k, each image is accompanied by 5 available ground-truth captions, with each caption utilized to calculate the CIDEr score. We perform APGD attacks with 100 iterations on each ground truth. After each attack, we calculate the CIDEr score and cease attacking samples that score below 10 for COCO or 2 for Flickr30k. This strategy is primarily employed because generating adversarial attacks on VLMs is extremely time-consuming and costly, allowing us to allocate our limited resources to more impactful attacks. For the VQA task, we apply a similar approach, but the score threshold is set to 0, and we utilize the 5 most common ground truths out of the 10 available.

## A.2 TARGETED ATTACK DETAILS

We use the following target sequences to perform targeted attacks:

- `A group of people are playing basketball.`
- `A group of people are flying in the air.`
- `The pizza on the table is very bad.`
- `An earthquake is about to happen.`
- `This patient needs the best medical care.`

If these targeted and even subtly provocative phrases are successfully embedded into adversarial images, they can subtly deceive users, thereby influencing their subsequent decisions. In addition to the quantitative evaluation presented in Section 4.2, we also observed that when the target text is closely related to the image content, the success rate of adversarial attacks is significantly high, indicating that images can easily mislead LVLMs. This presents a more dangerous scenario because when the target text is only weakly related to the image, users can more easily spot these erroneous outputs, thereby reducing their trust in the model. Conversely, when the model's output appears somewhat plausible in relation to the image content, users are more likely to trust the model's output.

## A.3 ZERO-SHOT EVALUATIONS

We evaluated the model's clean performance and robustness on a series of zero-shot image classification tasks. These datasets include CalTech (Griffin et al., 2007), StanfordCars (Krause et al., 2013), CIFAR10, CIFAR100 (Krizhevsky & Hinton, 2009), DTD (Cimpoi et al., 2014), EuroSAT (Helber et al., 2019), FGVC Aircrafts (Maji et al., 2013), Flowers (Nilsback & Zisserman, 2008), ImageNet-R (Hendrycks et al., 2021), ImageNet-Sketch (Wang et al., 2019), and PCAM (Veeling et al., 2018). The evaluation protocol is based on the *CLIP Benchmark*[1].

We assess the robustness by evaluating 1000 samples per dataset and reporting the clean accuracy for all samples. We utilize the first two attacks from AutoAttack (Croce & Hein, 2020), specifically, APGD with cross-entropy loss and APGD with targeted DLR loss, each with 100 iterations. Given that the DLR loss is applicable only to multi-class classification, we employ only the first attack on the binary dataset PCAM. We consider $\ell_\infty$-bounded threat models with radii $\epsilon = 4/255$ and evaluate the robustness on all datasets at a resolution of 224x224, except for CIFAR10, CIFAR100, and STL-10, which are evaluated at their original resolutions.

In Section 4.3, we only presented the performance of different CLIP versions on clean images and adversarial images with noise set to $\epsilon = 4/255$ due to space constraints. In Table 5, we show the evaluation results for an attack noise of $\epsilon = 2/255$. Humans can barely distinguish between images with $2/255$ noise and clean images, yet even such a small amount of noise causes the original CLIP model to nearly lose all its performance. This vulnerability is extremely critical. After adversarial

---

[1]https://github.com/LAION-AI/CLIP_benchmark

Table 5: **Evaluation of the clean performance and adversarial robustness with a noise $\epsilon = 2/255$ of different CLIP versions.**

| Eval. | Image Encoder | CalTech | Cars | CIFAR10 | CIFAR100 | DTD | EuroSAT | FGVC | Flowers | ImageNet-R | ImageNet-S | PCAM |
|---|---|---|---|---|---|---|---|---|---|---|---|---|
| $\epsilon = 2/255$ | CLIP | 0.0 | 0.0 | 0.0 | 0.0 | 0.0 | 0.0 | 0.0 | 0.0 | 0.0 | 0.1 | 0.0 |
| | TeCoA$^2$ | 70.2 | 22.2 | 63.7 | 35.0 | 27.0 | 12.8 | 5.8 | 27.6 | 58.8 | 45.2 | 40.0 |
| | FARE$^2$ | 73.0 | 26.0 | 60.3 | 35.6 | 26.7 | 6.2 | 5.9 | 31.2 | 56.5 | 38.3 | 41.9 |
| | AdPO$^2$ | 75.1 | 29.1 | 64.1 | 35.4 | 26.9 | 10.5 | 6.4 | 33.3 | 59.2 | 45.7 | 43.5 |
| | TeCoA$^4$ | 69.7 | 17.9 | 59.7 | 33.7 | 26.5 | 8.0 | 5.0 | 24.1 | 59.2 | 43.0 | 48.8 |
| | FARE$^4$ | 76.7 | 30.0 | 57.3 | 36.5 | 28.3 | 12.8 | 8.2 | 31.3 | 61.6 | 41.6 | 50.2 |
| | AdPO$^4$ | 78.1 | 32.5 | 64.2 | 36.1 | 27.4 | 13.9 | 9.3 | 34.2 | 62.4 | 42.5 | 51.3 |

training, multiple CLIP versions achieved noticeable adversarial robustness, but at the cost of some clean performance. Overall, AdPO had the least sacrifice in clean performance.

