# OpenReview forum: "AdPO: Enhancing the Adversarial Robustness of Large Vision-Language Models with Preference Optimization"
_ICLR.cc/2025/Conference — ICLR 2025 Conference Withdrawn Submission_

### Official Review · Reviewer_LrRQ · 2024-11-01

**Soundness:** 3
**Presentation:** 3
**Contribution:** 3
**Rating:** 5
**Confidence:** 3

**Summary:**

Existing methods enhance robustness through adversarial fine-tuning but often at the cost of degraded performance on clean inputs. This paper pioneers the reinterpretation of adversarial training as a preference optimization problem, aiming to strengthen the model’s preference for generating correct outputs on clean inputs while effectively rejecting misleading outputs for adversarial examples. AdPO achieves this by modifying only the image encoder, incorporating both preferred and adversarial image optimizations to improve robustness across clean and adversarial samples, preserving model performance on unaltered inputs.

**Strengths:**

- This study is the first to apply preference optimization in adversarial training, treating adversarial training as a preference-based optimization problem by integrating clean and adversarial images into the training process, enhancing robustness while maintaining performance on clean samples.

- The proposed combination of preferred image optimization and adversarial image optimization is innovative. Preferred image optimization uses an online approach to prompt the model directly, which alleviates the need for data annotation and helps mitigate distribution shifts, while adversarial image optimization uses context-aware outputs to handle adversarial samples. The joint optimization strategy enhances adversarial robustness while sustaining performance on clean data.

- Experimental results demonstrate AdPO's effectiveness and practical utility, showcasing significant performance gains across multiple datasets and tasks compared to baseline methods.

**Weaknesses:**

- The method section introduces preferred and adversarial image optimizations; however, the experimental section lacks a comparative analysis to showcase the specific benefits of these optimization methods.

- Joint optimization may add complexity, making training more challenging, particularly in terms of hyperparameter tuning and convergence. The paper could benefit from a discussion on the complexities and potential trade-offs involved in joint optimization.

- Although adversarial robustness is a core focus of this paper, the evaluation under the untargeted attack setting is limited to a single attack method. A more comprehensive evaluation would benefit from employing multiple attack types, including stronger attacks, to thoroughly assess the model's robustness. This approach would provide a clearer and more detailed understanding of AdPO's efficacy across a broader range of adversarial scenarios.

**Questions:**

See Weaknesses.

---

### Official Review · Reviewer_ZUp1 · 2024-11-01

**Soundness:** 2
**Presentation:** 3
**Contribution:** 2
**Rating:** 3
**Confidence:** 3

**Summary:**

In this paper a new adversarial learning based defense is proposed for LVLMs, the defense termed AdPO is  based on preference optimization (PO). Adversarial examples are generated and then used in the PO framework to train the model to be robust. The proposed approach achieves a better clean accuracy-robustness tradeoff as compared to the baselines. AdPO is tested on a couple of LVLMs across a variety of tasks.

**Strengths:**

- The idea of using preference optimization is new and interesting, also seems to work.
- For the tasks evaluated on - AdPO seems to yield similar or better robustness as competitors while attaining higher clean performance.
- The proposed method does not require labelled data and hence any image dataset can be used.

**Weaknesses:**

- Eqn. 4 is basically the loss as in FARE (see Eqn. 3 and discussion in section 3.3 in [1]) - and this discussion is missing from the text. This seems misleading, since this equation does the heavy-lifting in generating adversarial examples that make AdPO robust.

- The fact that AdPO leads to more robust models is not surprising since the method finetunes the whole LVLM (LLM is frozed but the loss/gradients are propogated through) unlike FARE and TECOA which finetune just the vision encoder. The comparison's thus seem not fair as the settings are different. There should have been standardization of FARE and TEcoA at the same level of compute-time (could be done with more epochs/more data etc for FARE/TECOA ) as AdPO.

- Overall time complexity of the method seems very high (relative to baselines) and is not described in the paper. Given the loss is calculated over the whole LVLM, and there is online description generation of the image within the method - it seems significantly higher that TECOA or FARE.

- Table-2 does not add help AdPO's efficacy much, since the number of samples is small and the difference between FARE$^2$ and AdPO$^2$ is only 2%, and there is no difference for larger radii.

- In Figure3, AdPO does not seem better than FARE, for instance:
"FARE: A young boy is walking on a grassy field." is subjectively slightly better description than "AdPO: A young boy is playing
with a soccer ball in a field." since the boy is walking while holding the ball and not playing with it.

- Lacks other important evaluations: other important aspects of LVLMs like hallucination, robustness to jailbreaks should have been tested with AdPO, does the proposed method also help in these cases? This is important as making models robust should not lead to substantial degradation in performance in other regards - else the model is not usable as before.

[1] Schlarmann, Christian, et al. "Robust CLIP: Unsupervised Adversarial Fine-Tuning of Vision Embeddings for Robust Large Vision-Language Models." Forty-first International Conference on Machine Learning.

**Questions:**

- In Table-3 for some setting TeCOA performs better than ADPO - is there any intuition why this would be the case?
- How were the target captions for the targeted attack selected?

---

### Official Review · Reviewer_V42j · 2024-11-04

**Soundness:** 3
**Presentation:** 3
**Contribution:** 3
**Rating:** 5
**Confidence:** 4

**Summary:**

This paper proposes AdPO, a method for adversarial training based on preference optimization, which improves the robustness of LVLM.

**Strengths:**

1. This paper is clearly written.

2. The paper is well evaluated on different models (OpenFlamingo and LLaVA) and datasets (COCO, Flickr30k, TextVQA, VQAv2).

3. In the authors' evaluation setup, AdPO shows non-trivial improvements compared to the baseline approach.

**Weaknesses:**

1. The attack strengths used in adversarial training and evaluation were only 2 and 4; does AdPO still perform better than other baselines at larger attack strengths (e.g., 8)? Robustness evaluation under other attack methods, e.g., unconstrained strong CW attack, adversarial attack with smoothing adversarial images, is also expected.

2. Complexity analysis of AdPO. What is the amount of adversarial training data and the computational cost of adversarial training between AdPO and TeCoA and FARE? If AdPO uses less data or is more efficient, then this is another advantage of AdPO. If AdPO uses more data or requires more computational cost, then a fair comparison should be made.

3. Lack of performance comparison with direct adversarial training. While the cost of adversarial training is high, 32 A100 GPUs can support this training, would AdPO work better than direct adversarial training against LVLM?

4. The clean COCO performance of AdPO2 in Table 1 even exceeds that of CLIP, why might this be?

5. Despite what is stated in line 530, I'm still curious why AdPO is only done on CLIP, if end-to-end AdPO would lead to better improvements.

6. Can you show the results after scaling up? AdPO should be more efficient than adversarial training, can using more high quality DPO data further improve the performance?

7. Many variants of DPO exist nowadays and it is expected that the authors discuss or compare with these DPO variants to further demonstrate the effectiveness of AdPO.

**Questions:**

See weaknesses.

---

### Official Review · Reviewer_huwk · 2024-11-04

**Soundness:** 2
**Presentation:** 3
**Contribution:** 2
**Rating:** 3
**Confidence:** 4

**Summary:**

This paper introduces a new defense method for LVLM attacks. Different from previous finetuning-based defenses, this paper proposes a new adversarial training strategy with preference optimization methods. It generates the sample pairs to train the model. Experiments are conducted on two LVLM models.

**Strengths:**

1. This paper is well-organized and easy to read.

2. The motivation is straightforward and reasonable.

**Weaknesses:**

1. The novelty is weak. The proposed method just applies previous DPO method into the LVLM scenarios. The main contribution is to generate prefer and non-prefer pairs for training. All the components are not new. Therefore, the contribution of this paper to the field of LVLM adversarial defense is weak.

2. Missing many related references of LVLM attacks and defenses. There are many latest LVLM attack and defense papers released on Arxiv. There are also many survey papers that summarize and discuss existing LVLM attacks and defenses. However, this paper does not cite, summarize, and discuss these existing works. The author only introduces part of it, thus the discussion is not comprehensive and the comparison is not fair.

3. Experiments are not convincing. First, the authors just simply conduct experiments on two simple LVLM models. More experiments on the general LVLM models (like GPT4, BLIP2) are required. Otherwise, it cannot validate the generalization of the proposed method. Second, the authors just implement two defenses method for comparison. This is not enough. At least 4-5 defense baselines are needed.

4. Since this paper aims to improve the adversarial robustness of existing LVLM models, it should provide in-depth analysis of the defense performance to existing LVLM attackers. However, the authors do not provide any analysis.

5. No complexity and efficiency discussion. It is necessary to provide corresponding analysis compared to existing defense methods.

**Questions:**

1. The novelty is weak. The proposed method just applies previous DPO method into the LVLM scenarios. The main contribution is to generate prefer and non-prefer pairs for training. All the components are not new. Therefore, the contribution of this paper to the field of LVLM adversarial defense is weak.

2. Missing many related references of LVLM attacks and defenses. There are many latest LVLM attack and defense papers released on Arxiv. There are also many survey papers that summarize and discuss existing LVLM attacks and defenses. However, this paper does not cite, summarize, and discuss these existing works. The author only introduces part of it, thus the discussion is not comprehensive and the comparison is not fair.

3. Experiments are not convincing. First, the authors just simply conduct experiments on two simple LVLM models. More experiments on the general LVLM models (like GPT4, BLIP2) are required. Otherwise, it cannot validate the generalization of the proposed method. Second, the authors just implement two defenses method for comparison. This is not enough. At least 4-5 defense baselines are needed.

4. Since this paper aims to improve the adversarial robustness of existing LVLM models, it should provide in-depth analysis of the defense performance to existing LVLM attackers. However, the authors do not provide any analysis.

5. No complexity and efficiency discussion. It is necessary to provide corresponding analysis compared to existing defense methods.

---

### Note · Authors · 2024-11-15

**Comment:**

We have decided to withdraw the paper, but we would like to express our gratitude for the constructive feedback in the review comments. We will further clarify any potential misunderstandings in future versions.

**Withdrawal Confirmation:**

I have read and agree with the venue's withdrawal policy on behalf of myself and my co-authors.